# Preparation and pH Controlled Release of Fe_3_O_4_/Anthocyanin Magnetic Biocomposites

**DOI:** 10.3390/polym11122077

**Published:** 2019-12-12

**Authors:** Xizhi Jiang, Qingbao Guan, Min Feng, Mengyang Wang, Nina Yan, Min Wang, Lei Xu, Zhongzheng Gui

**Affiliations:** 1School of Biotechnology, Jiangsu University of Science and Technology, Zhenjiang 212018, China; jiangxizhi@jaas.ac.cn; 2Institute of Agricultural Facilities and Equipment, Jiangsu Academy of Agricultural Sciences, Nanjing 210014, China; fengmin8156@163.com (M.F.); wmy19971022@163.com (M.W.); yannina@jaas.ac.cn (N.Y.); wangminwj@163.com (M.W.); 3Key Laboratory for Protected Agricultural Engineering in the Middle and Lower Reaches of Yangtze River, Ministry of Agriculture and Rural Affairs, Nanjing 210014, China; 4State Key Laboratory for Modification of Chemical Fibers and Polymer Materials, International Joint Laboratory for Advanced Fiber and Low-dimension Materials, College of Materials Science and Engineering, Donghua University, Shanghai 201620, China; qbguan@dhu.edu.cn

**Keywords:** magnetic biocomposite, anthocyanin, Fe_3_O_4_, storage, release

## Abstract

Anthocyanins are a class of antioxidants extracted from plants, with a variety of biochemical and pharmacological properties. However, the wide and effective applications of anthocyanins have been limited by their relatively low stability and bioavailability. In order to expand the application of anthocyanins, Fe_3_O_4_/anthocyanin magnetic biocomposite was fabricated for the storage and release of anthocyanin in this work. The magnetic biocomposite of Fe_3_O_4_ magnetic nanoparticle-loaded anthocyanin was prepared through physical intermolecular adsorption or covalent cross-linking. Scanning electron microscopy (SEM), Dynamic light scattering (DLS), Fourier-transform infrared spectroscopy (FTIR), X-ray diffractometry (XRD) and thermal analysis were used to characterize the biocomposite. In addition, the anthocyanin releasing experiments were performed. The optimized condition for the Fe_3_O_4_/anthocyanin magnetic biocomposite preparation was determined to be at 60 °C for 20 h in weak alkaline solution. The smooth surface of biocomposite from SEM suggested that anthocyanin was coated on the surface of the Fe_3_O_4_ particles successfully. The average size of the Fe_3_O_4_/anthocyanin magnetic biocomposite was about 222 nm. Under acidic conditions, the magnetic biocomposite solids could be repeatable released anthocyanin, with the same chemical structure as the anthocyanin before compounding. Therefore, anthocyanin can be effectively adsorbed and released by this magnetic biocomposite. Overall, this work shows that Fe_3_O_4_/anthocyanin magnetic biocomposite has great potential for future applications as a drug storage and delivery nanoplatform that is adaptable to medical, food and sensing.

## 1. Introduction

Anthocyanins are hydrophilic and water-soluble polyphenolic plant pigments and metabolites, which are one of the important antioxidants. They possess a wide range of pharmacological properties, such as antioxidant, antiaging, anti-inflammatory, antimicrobial and anti-cancer [1,2,3,4]. 

Current researches have demonstrated the strong antimicrobial effects of anthocyanins, e.g., inhibiting the growth of *Listeria monocytogenes*, *Staphylococcus aureus*, *Salmonella enteritidis* and *Vibrio parahaemolyticus* [5]. In addition, anthocyanins can lower blood glucose levels by protecting b-cells, improving insulin resistance, increasing insulin secretion, improving liver function and inhibiting carbohydrate hydrolyzing enzymes [6]. Furthermore, anthocyanins have the ability to induce HepG-2 cell apoptosis [7] and inhibit B16-F1 cell migration and Human umbilical vein endothelial cells (HUVECs) tube formation [8], showing strong anticancer activity. Therefore, anthocyanins have a great potential in various fields, such as the pharmaceutical and food industries. However, the relatively low stability and bioavailability of anthocyanins are the primary barrier to limit the wide and effective applications [9,10,11], raising up the importance of reducing the degradation of anthocyanins and controlling release. So far, encapsulation [12,13,14,15], composite [16,17,18,19] and group substitution [20,21,22,23,24] are the three main means used for anthocyanin stabilization. Some studies have been carried out on the adsorption of pigments by Fe_3_O_4_ [25] and the stabilization of Fe^3+^ [26,27]. The development of magnetic delivery systems based on the surface modified by polymers can combine and adsorb pigment molecules, showing a high effectivity.

Magnetic nanoparticles exhibit superparamagnetism and magnetic field response, giving broad application prospects in the fields of biology and biomedicine [28,29]. Several studies have shown that magnetic nanoparticles exhibit extremely low toxicity to the human body, as a result of the degradation and release of Fe^3+^ ions that participate in iron metabolism in the liver and spleen [30]. Magnetic biocomposite with nanoparticles as carriers is a promising method for controlling drug stability and the release of drug targeting. Hosseini et al. [31] have suggested that the drug-loaded, magnetic Fe_3_O_4_ nanoparticles showed excellent pH responsibility in vitro, and large amounts of doxorubicin releases occurred at lower pH. Rubia et al. [30] developed new biomimetic magnetite nanoparticles (Fe_3_O_4_) functionalized with doxorubicin, which is positively charged at pH 7.4, and also allows the stability of the doxorubicin surface bond and doxorubicin release to be pH-dependent and governed by electrostatic interactions. The above research work indicates that magnetic Fe_3_O_4_ nanoparticles could be useful as the potential nanodrug carriers with a drug adsorption release controlled by changes in local pH values. However, the studies on the adsorption of anthocyanin by Fe_3_O_4_ nanoparticles have not been reported.

In this paper, we proposed an innovative methodology for the preparation of anthocyanin storage and release nanoplatforms, based on magnetite Fe_3_O_4_ nanoparticles modified by amino polymers. The magnetic Fe_3_O_4_ nanoparticles were prepared through the hydrothermal synthesis method. Then cyanidin-3-*O*-glucoside standard was used as a natural pigment to investigate the effect of different mass ratios, reaction times, pH values and reaction temperatures on Fe_3_O_4_/anthocyanin magnetic biocomposite. A series of performance characterizations were carried out to evaluate the combination of anthocyanin and Fe_3_O_4_. The release tests were finally applied to obtain anthocyanin by varying the solutions, pH values and release times. The mechanism was also explored on the released anthocyanin.

## 2. Materials and Methods

### 2.1. Materials

The anthocyanin (cyanidin-3-*O*-glucoside standard) with a purity of 98.15% was purchased from Shanghai Macklin Biochemical Co., Ltd., Shanghai, China. Methanol (chromatographic grade) was purchased from Merck, Germany. Ferric chloride hexahydrate (FeCl_3_·6H_2_O), trisodium citrate dihydrate (Na_3_C_6_H_5_O_7_·2H_2_O), urea and polyacrylamide (PAM, *Mn* = 3,000,000 g·mol^−1^) were obtained from Shanghai Hushi Laboratory Equipment Co., Ltd., Shanghai, China. Ammonia solution (NH_3_·H_2_O, 25%), as well as the solvents including phosphoric acid and ethanol were provided by local suppliers and used as received. All reagents unless stated were analytical grade. Deionized water was used in all experiments.

### 2.2. Preparation of Magnetic Fe_3_O_4_ Nanoparticles

The magnetic Fe_3_O_4_ nanoparticles were prepared via the hydrothermal synthesis method. 0.54 g FeCl_3_·6H_2_O was first dissolved in 40 mL water to form a clear solution. Then 1.18 g Na_3_C_6_H_5_O_7_·2H_2_O and 0.36 g urea were added into the solution. The mixture was kept stirred until the solid dissolved completely at room temperature. 0.3 g polyacrylamide was then added into the solution and the mixture was stirred for 1 h at the speed of 700 rpm. The mixture was subsequently transferred into a Teflon®-lined, stainless-steel autoclave maintained at 200 °C for 12 h. The black products were obtained and then washed several times with ethanol and deionized water, followed with being dried in a vacuum oven [32].

### 2.3. Preparation of Fe_3_O_4_/Anthocyanin Magnetic Biocomposite

Anthocyanin was dissolved in a given volume of methanol. In order to remove the hydrogen of the phenolic groups, NH_3_·H_2_O was then added until the pH of the solution was kept between 7 and 8 [25]. Ultrasound-treated Fe_3_O_4_ methanol solution was injected into the anthocyanin mixture. The mass proportions of anthocyanin to Fe_3_O_4_ were set as 5:1, 1:1, 1:5, 1:10, 1:20, 1:40 and 1:80. Then to be further refined, the mass proportions of anthocyanin to Fe_3_O_4_ were 1:25, 1:30, 1:35, 1:40, 1:45, 1:50, 1:55, 1:60, 1:65 and 1:70. These 16 proportions of Fe_3_O_4_ and anthocyanin mixtures were oscillated at 60 °C for 20 h. After that, the supernatant was separated, and then the solid powder of biocomposite can be obtained by drying in a vacuum oven overnight. The Fe_3_O_4_/anthocyanin magnetic biocomposites were prepared with different times, pH values and temperatures.

### 2.4. Release Research

The Fe_3_O_4_/anthocyanin magnetic biocomposite solids were resuspended in methanol, deionized water and ethanol. The pH values of each biocomposite suspension (160 mM) were adjusted to 1.0, 1.6, 2.0 and 3.0, respectively. The release research was performed by oscillating suspension for 5 to 10 s at room temperature. After centrifugation, absorbance of the released anthocyanin from the samples were measured by a UV spectrophotometer at 520 nm.

### 2.5. Characterizations

The UV–vis spectra of the anthocyanin solutions were obtained by an ultraviolet-visible spectrophotometer (UV-3200PC, Mapada, Shanghai, China) at 520 nm. The morphologies of the samples before and after compounding were examined on cover slips with an EVO-LS10 scanning electron microscope (SEM, Zeiss, Jena, Germany) at the accelerating voltage of 10.00 kV. In order to improve the conductivity of the sample, a thin layer of platinum–palladium alloy was first deposited in vacuum. Dynamic light scattering (DLS) measurement was also performed with a nanoparticle size and zeta potential analyzer (NICOMP Z3000, PSS, Santa Barbara, CA, USA) for determining the sizes and zeta potentials of the nanoparticles dispersed in water. Fourier transformation infrared spectra (FTIR) of the samples before and after compounding were characterized by a Nicolet iS50 infrared spectrometer (Thermo Fisher Scientific, Waltham, MA, USA) under the attenuated total reflection mode (ATR). X-ray diffraction (XRD, D8 Advance, Bruker, Karlsruhe, Germany) patterns of the nanoparticles were obtained from a wide-angle diffractometer with Cu Kα radiation (λ = 1.5418 Å) at a generator voltage of 20 kV and a generator current of 5 mA. The scanning speed and the step were 4°/min and 0.02°, respectively. The thermal stability of the nanoparticles was tested by a thermogravimetric analysis (TG/DTA7200, SII, Tokyo, Japan) and a differential scanning calorimetry (DSC Q20, TA, New Castle, DE, USA) machine. The analysis was performed under nitrogen atmosphere with a heating rate of 20 °C min^−1^ from 30 to 600 °C. After the dilution of samples in deionized water, anthocyanin was quantified by using high performance liquid chromatography (HPLC 1260II, Agilent, Santa Clara, CA, USA), equipped with a quaternary pump, an autosampler and a DAD detector. A reverse phase column (Poroshell 120 EC-C18, 4.0 μm, 4.6 mm × 150 mm) was used at 20 °C for the detection of the anthocyanins. The flow rate was 0.8 mL/min and the injection volume was 2.0 μL. 

Mobile phases consisted of 1% formic acid in water (A) and 1% formic acid in acetonitrile (B). The test was carried out under the following conditions: 0.00–2.50 min, linear gradient from 8% to 12% B; 2.50–5.00 min, linear gradient from 12% to 18% B; 5.00–10.10 min, linear gradient from 18% to 20% B; 10.10–12.00 min, linear gradient from 20% to 80% B; 12.00–14.00 min, 80% B; 14.00–14.10 min, linear gradient from 80% to 8% B; 14.10–18.00 min, 8% B; and finally, washing and re-equilibration of the column prior to the next injection. All samples were kept at 4 °C during the analysis. Anthocyanin was monitored at 520 nm. The relative molecular mass of anthocyanin was measured using the LC-MS (LC 1260 MS G6420A, Agilent, Santa Clara, CA, USA) to identify. The MS parameters were as follows: EIS of positive mode; capillary voltage, 4000 V; gas flow rate, 10 L/min; gas temperature, 350 °C; and scan range, 100–700 *m*/*z*. Prior to injecting, each sample was filtered through a 0.22 μm syringe filter.

## 3. Results

### 3.1. Preparation of Fe_3_O_4_/Anthocyanin Magnetic Biocomposite

A magnetic delivery system which could be stabilized by physical intermolecular or covalent cross-linking by altering temperatures or pH values is based on the surface modified by polymers [25]. The schematic illustration of the preparation route for the Fe_3_O_4_/anthocyanin magnetic biocomposite was depicted in Figure 1. An anthocyanin linkage way with metal oxide to form the metal oxide–anthocyanin complex was proposed in this work. The complex was first prepared in weak alkaline solution with the temperature of 60 °C for 20 h, and solid –liquid separation was carried out under magnetic field to collect the solids. Eventually, an acidic solution was added to release anthocyanin.

The color changes of solutions during the synthesis of Fe_3_O_4_/anthocyanin magnetic biocomposites were shown in Appendix A. As shown in Appendix A, the mass portion of Fe_3_O_4_ has a great influence on the synthesis of magnetic biocomposites. As depicted in Appendix A, all of the solutions appeared in red when the mass ratio of anthocyanin to Fe_3_O_4_ decreased from 5:1 to 1:20, which represents the excess of the anthocyanin. While the mass portion of Fe_3_O_4_ increased, the red became shallow gradually. The colors of solutions changed from pale red to transparent when the mass ratio decreased from 1:20 to 1:40, indicating that almost all the anthocyanin has reacted with Fe_3_O_4_. When the mass ratio further decreased from 1:40 to 1:160, the solutions remained transparent, which indicated that the Fe_3_O_4_ was excessive. 

As demonstrated in Appendix A, the solution color changed from pale red (1:25~1:40) to pale blue (1:45~1:50) to yellowish (1:55~1:70), displaying a process form the excess of anthocyanin to the micro-dissolution of excess Fe_3_O_4_.

Appendix A showed the effect of different reaction time on the color of the solution. The complex solutions were alkaline at the beginning of the reaction, and the anthocyanin was blue–purple. With the prolongation of the reaction time, the color changed from blue–purple to yellow, and finally to transparent [26,33]. As the reaction time prolonged to 20 h, the solution was colorless, as shown in Appendix A. When the reaction time further extended to 22 h, the solution turned slightly yellow due to the saturation of the Fe_3_O_4_/anthocyanin magnetic biocomposites. The effect of different pH values on the color of the solutions was exhibited in Appendix A. The solution appeared light black when the pH was no more than 7.0. The solution became clear when the pH was more than 7.0, and it became colorless and transparent at pH 8.0, as shown in Appendix A. The solution gradually turned yellow with the pH value increased, caused by excessive ammonia water. Appendix A showed the relationship between temperature and the solution color, and the solution was colorless, clear and transparent at 60 °C. 

The ultraviolet–visible absorbance of Fe_3_O_4_/anthocyanin magnetic biocomposites measured at 520 nm were depicted in Figure 2. Figure 2a,b showed the influence of the different mass ratios of the two substances on the UV-vis absorbance. It was found that the absorbance decreased with the mass ratio from 5:1 to 1:40, while the absorbance increased when the mass ratio of anthocyanin to Fe_3_O_4_ decreased from 1:40 to 1:160. Figure 2b showed that the absorbance of the mass ratio from 1:50 to 1:70 kept below 0.04. As shown in Figure 2c, the absorbance decreased with the reaction time from 0 to 20 h. However, when the time further extended to 22 h, the absorbance increased slightly due to a slight dissolution or the reverse reaction of the complex. Figure 2d showed that the minimum absorbance value was obtained at pH 8.0. Regulating the solution to alkaline aimed to remove hydrogen ions from the phenolic group and to promote the reaction between the hydroxyl groups of anthocyanin and the hydroxyl groups of Fe_3_O_4_ [25]. However, excessive –OH due to continuous addition of ammonia solution was not conducive to the reaction. As demonstrated in Figure 2e, the absorbance decreased linearly with the increase of temperature. However, higher temperature would promote the degradation of anthocyanin. Therefore, 60 °C is chosen to be the most suitable temperature.

### 3.2. Properties of Fe_3_O_4_/Anthocyanin Magnetic Biocomposites

The morphologies of Fe_3_O_4_ nanoparticles and Fe_3_O_4_/anthocyanin magnetic biocomposites were characterized by SEM. As shown in Figure 3a, Fe_3_O_4_ nanoparticles prepared by the hydrothermal method were well monodispersed with a diameter of approximately 200~300 nm. The surface of Fe_3_O_4_ nanoparticles was rough and the structures of the nanoparticles were porous. Compared with the Fe_3_O_4_ nanoparticles shown in Figure 3a, the surface of Fe_3_O_4_/anthocyanin magnetic biocomposites became smooth, which indicated that anthocyanin has been coated on the surface of the Fe_3_O_4_ particles (Figure 3b).

The zeta potentials and particle sizes of anthocyanin, Fe_3_O_4_ nanoparticles and Fe_3_O_4_/anthocyanin composite were summarized in Table 1. According to the zeta potential results, the anthocyanin solution was positively-charged, while the Fe_3_O_4_ solution was negatively-charged. Therefore, there was an electrostatic adsorption between anthocyanin and Fe_3_O_4_ particles, which was propitious to the combination of the two materials to form Fe_3_O_4_/anthocyanin magnetic biocomposites. Compared with Fe_3_O_4_ nanoparticles, the zeta potential and particle size both increased after the combination of Fe_3_O_4_ particles and anthocyanin.

Particle size distributions of Fe_3_O_4_ and Fe_3_O_4_/anthocyanin magnetic biocomposites were shown in Figure 3c,d. Each sample was tested three times. It could be seen that the particle size distributions of Fe_3_O_4_ and Fe_3_O_4_/anthocyanin magnetic biocomposites were stable. The particle size distributions of Fe_3_O_4_ ranged from 30 to 1100 nm, and were concentrated between 50 and 300 nm, while that of Fe_3_O_4_/anthocyanin magnetic biocomposites ranged from 30 to 1000 nm and were concentrated between 100 and 400 nm. Based on the measurement, the particle size of Fe_3_O_4_/anthocyanin magnetic biocomposites was larger than that of Fe_3_O_4_.

FTIR spectra of anthocyanin, Fe_3_O_4_ nanoparticles and Fe_3_O_4_/anthocyanin composites were depicted in Figure 4a. The peaks of anthocyanin at 3270, 2980, 1640, 1420, 1080 and 1040 cm^−1^ represented the –OH functional groups, symmetric –CH stretching vibrations modes of methyl groups, C=O of the benzopyran aromatic ring vibration, C=C aromatic ring stretching vibrations, C–H aromatic ring deformation, and C–O vibration, respectively [19,34]. Fe_3_O_4_ nanoparticles exhibited a weak band at 3280 cm^−1^ which was attributed to free vibration of O–H stretch as the material contains a hydroxyl group. Furthermore, the spectrum of Fe_3_O_4_ showed two clear absorption bands respectively at 1550 cm^−1^ and 1400 cm^−1^ corresponding to O–H and C–N absorption peaks [25]. The new characteristic peaks from 1000 cm^−1^ to 1300 cm^−1^ on the curve of Fe_3_O_4_/anthocyanin composite caused by C–O stretch of anthocyanin proved that anthocyanin has been synthesized with Fe_3_O_4_. It was observed that the absorption bands in O–H at 3280 cm^−1^ and 1550 cm^−1^ were nearly unchanged due to the balance between the introduced hydroxyl groups of anthocyanin and the consumed hydroxyl groups.

Figure 4b exhibited the X-ray diffraction pattern of the Fe_3_O_4_ and Fe_3_O_4_/anthocyanin magnetic biocomposites. The diffraction peaks at 30.1°, 35.6°, 43.2°, 53.5°, 57.1° and 62.7° shown in both XRD patterns corresponded to the indices of crystal face of (220), (311), (400), (422), (511) and (440), respectively [35], which was in agreement with standard (JCPDS Card No. 1-1111). The characteristic peaks of Fe_3_O_4_/anthocyanin and Fe_3_O_4_ nanoparticles were consistent, proving the crystal structures of Fe_3_O_4_ remained unchanged after compounding with anthocyanin.

The thermo-gravimetric analysis and differential scanning calorimeter were used to determine the mass loss and heat flow of Fe_3_O_4_ nanoparticles in comparison with Fe_3_O_4_/anthocyanin magnetic biocomposites. The TGA curves and the DSC curves of samples were given in Figure 4c,d respectively. The results showed that weight loss at 125 °C with an endothermic peak was mainly due to the loss of adsorbed water molecules on the surface of nanoparticles. 

However, this endothermic peak of Fe_3_O_4_/anthocyanin magnetic biocomposites was wider than that of Fe_3_O_4_, owing to more heat demand of the composites. Fe_3_O_4_ and Fe_3_O_4_/anthocyanin samples have a 2.6% and a 3.3% weight loss from 125 to 275 °C, respectively, corresponding to the degradation of different organic groups [31]. The weight losses of both samples were 3.5% in the temperature range of 275 to 385 °C, with an endothermic peak at 332 °C due to the partial changes of the crystal form of Fe_3_O_4_ [35]. The weight losses of both samples were the same with the temperature increasing from 385 °C to 600 °C. However, the wide endothermic peak of Fe_3_O_4_/anthocyanin was at 432 °C, which appeared 30 °C ahead of the Fe_3_O_4_ peak, due to the changes in the Fe_3_O_4_ molecules at this stage.

### 3.3. Release of Anthocyanins from Fe_3_O_4_/Anthocyanin Magnetic Biocomposites

The release results of anthocyanin under different conditions were depicted in Appendix A. The effects of solutions, pH values and release times on anthocyanin release were discussed. Appendix A showed the anthocyanin (I) before compounding with Fe_3_O_4_, the complex (II) of anthocyanin and Fe_3_O_4_ after compounding, solid complex (III) before releasing and supernatant (IV) before releasing, respectively. Appendix A were the results of anthocyanin released from solid complex using methanol, water and ethanol with different pH values. It was found that anthocyanin could be released by adding any of these three solutions. Acidic conditions can promote the release of anthocyanin. The solution color got darker when the pH value decreased, and meanwhile the release percentage increased. However, when released in deionized water solution, the solution turned black at the pH of 2.0 and above, due to the better hydrophilia and dissolution of the complex. Appendix A show the secondary release of anthocyanin from methanol and water solutions with different pH values. It was observed that multiple releases (i.e., release more than once in one solution) could be achieved, although the release effect was not as good as the first release.

The primary release percentages of anthocyanin were shown in Figure 5a. The maximum release percentages in methanol, water and ethanol were 60.9%, 54.1% and 22.2%, respectively, and the release percentage in methanol was the highest. When the pH value was 1.6, which was equal to the pH value of anthocyanin before compounding with Fe_3_O_4_, the release percentage in water was 21.1%, higher than that in methanol (17.9%) and ethanol (9.4%). When pH was equal to or higher than 2.0, the release percentages in methanol and ethanol were nearly the same, but the release percentage in water was higher. Combined with the analysis to Appendix A, the dissolution of the complex led to the solution with black color. The secondary release percentages of anthocyanin were exhibited in Figure 5b. The release percentage in methanol was found to be substantially higher than that in water. But the maximum secondary release percentage was 14.2%, which was lower than the primary release percentage. When the pH was equal to or higher than 2.0, the dissolution was still observed in water, leading to the higher tested value. Release percentages of anthocyanin from Fe_3_O_4_/anthocyanin magnetic biocomposites at different temperatures were also given in Appendix A. As the temperature changed from 30~50 °C, the release percentages remained around 30%, indicating that the magnetic biocomposites could be released stably at 30~50 °C.

Figure 5c showed the FTIR spectra of released anthocyanin in methanol solutions with different pH values. When the pH was 3.0, the infrared spectrum was almost the same with methanol, indicating nearly no release of anthocyanin. When the pH was 2.0, 1.6 and 1.0, the peaks at 3270, 2980, 1400 and 1020 cm^−1^ represent the –OH groups, symmetric –CH groups, –OH groups, and C–O groups, respectively. When the pH decreased from 3.0 to 1.0, the O–H peak gradually shifted from 3310 cm^−1^ to 3270 cm^−1^, indicating the hydroxyl groups of methanol gradually changed into the hydroxyl groups of anthocyanin. Figure 5d showed the FTIR spectra of released anthocyanin when methanol, water and ethanol were used as solvents, respectively. The groups of the released anthocyanin were affected by the solvents.

The anthocyanin before compounding and after releasing were also analyzed by HPLC and HPLC-MS to verify the chemical structures, and the results were shown in Figure 6. The structure of cyanidin-3-O-glucoside standard was shown in Appendix A. There was only one adsorption peak at 520 nm in the HPLC profile (Figure 6b), and the corresponding retention time was 6.286 min, which was quite similar to the retention time of the standard spectrum of cyanidin-3-*O*-glucoside (6.283 min in Figure 6a). Besides, the ions at the mass-to-charge ratio (*m*/*z*) 449 (Figure 6c,d) in the positive ionization mode were tested, which indicated that the released anthocyanin mainly consisted of cyanidin-3-*O*-glucoside and some other derivatives [36,37,38].

## 4. Conclusions

In this paper, an innovative strategy for the preparation of Fe_3_O_4_/anthocyanin magnetic biocomposites was proposed. Anthocyanin could be controlled-released from the biocomposites by pH values. With the introduction of the hydroxyl and electrostatic adsorption charges, the Fe_3_O_4_/anthocyanin magnetic biocomposites were supposed to be coordinated on the physical intermolecular or covalent cross-linking between Fe_3_O_4_ and anthocyanin. This strategy provides a feasible method for preparing Fe_3_O_4_/anthocyanin magnetic biocomposites with designated chemical groups. Magnetic biocomposites have an average size about 222 nm. The primary release percentage of anthocyanin in acid methanol was 60.9%. The preparation method of this work is simple, and the prepared biocomposites have good biocompatibility and could be controlled-released. The biocomposites could be released at the target position by magnetic guide. Therefore, the methodology of Fe_3_O_4_/anthocyanin magnetic biocomposites presented in this work provides an effective way to combine, adsorb and release anthocyanin, which has great potential value in the applications of medical, food and sensing.

## Figures and Tables

**Figure 1 polymers-11-02077-f001:**
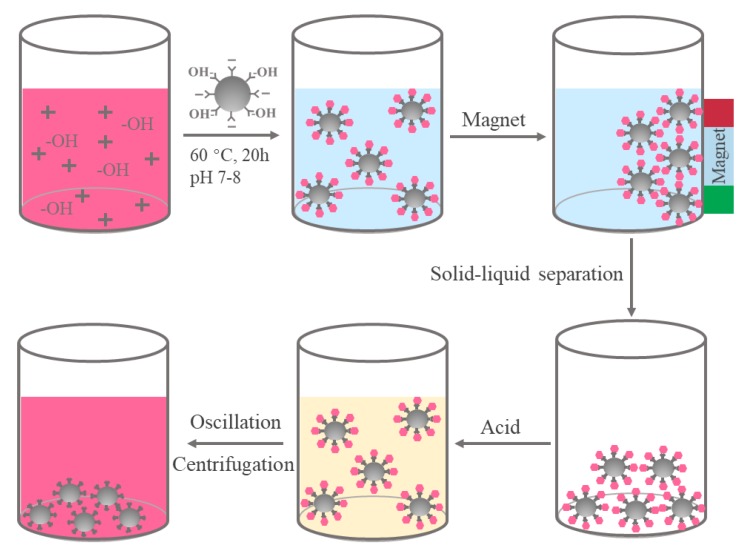
The schematic diagram for the fabrication of magnetic biocomposites through anthocyanin and Fe_3_O_4_.

**Figure 2 polymers-11-02077-f002:**
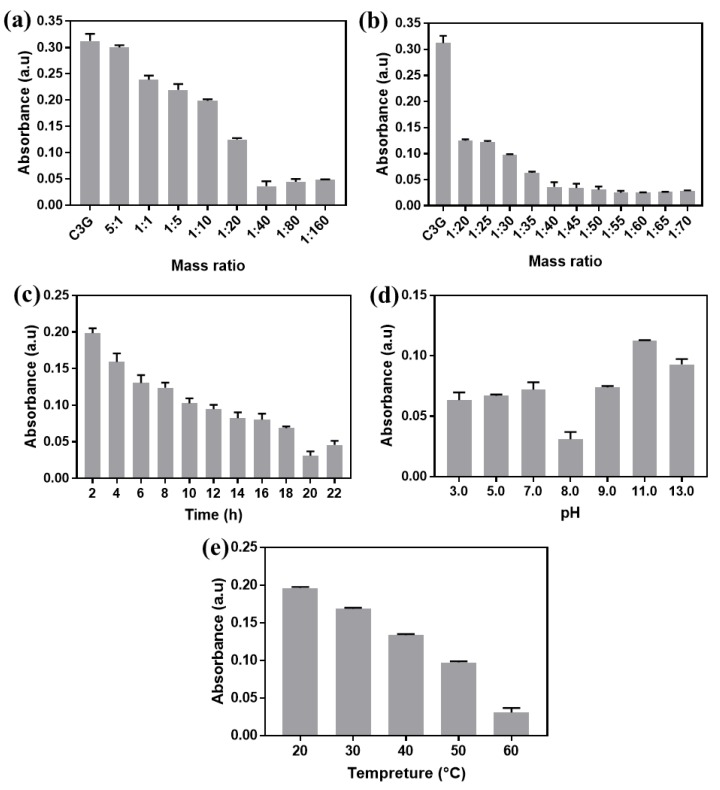
The UV–vis absorbances of Fe_3_O_4_/anthocyanin magnetic biocomposites with (**a**,**b**) different anthocyanin and Fe_3_O_4_ mass ratios; (**c**) different reaction times; (**d**) different pH; and (**e**) different reaction temperatures.

**Figure 3 polymers-11-02077-f003:**
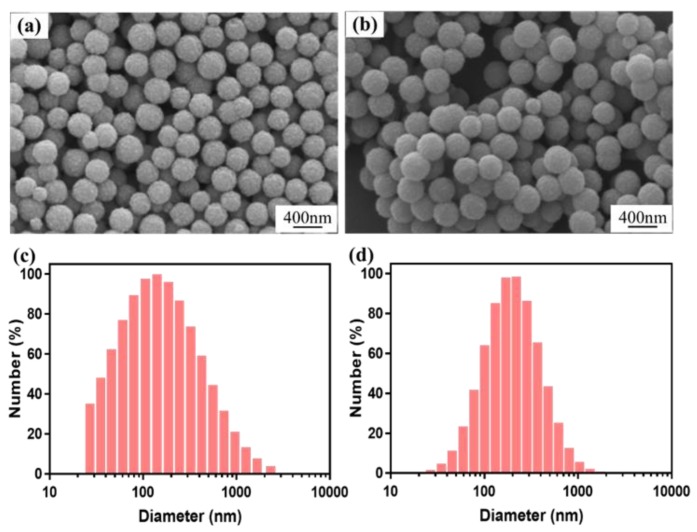
Scanning electron microscope (SEM) images of (**a**) Fe_3_O_4_ nanoparticles and (**b**) Fe_3_O_4_/anthocyanin magnetic biocomposites. Particle size distributions of Fe_3_O_4_ (**c**) and Fe_3_O_4_/anthocyanin composite (**d**) by dynamic light scattering (DLS) analysis.

**Figure 4 polymers-11-02077-f004:**
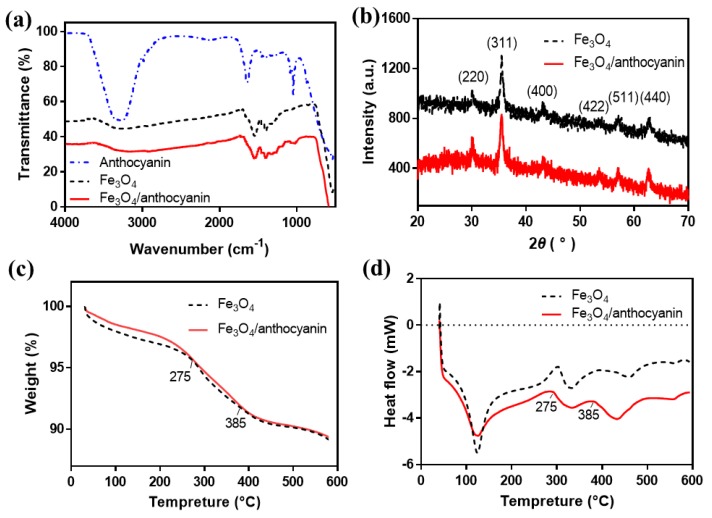
(**a**) Fourier transformation infrared spectra (FTIR) spectra of anthocyanin, Fe_3_O_4_ and Fe_3_O_4_/anthocyanin composites; (**b**) X-ray diffractometry (XRD) patterns, (**c**) thermogravimetric analysis (TGA) curves and (**d**) differential scanning calorimetry (DSC) curves of Fe_3_O_4_ and Fe_3_O_4_/anthocyanin composites.

**Figure 5 polymers-11-02077-f005:**
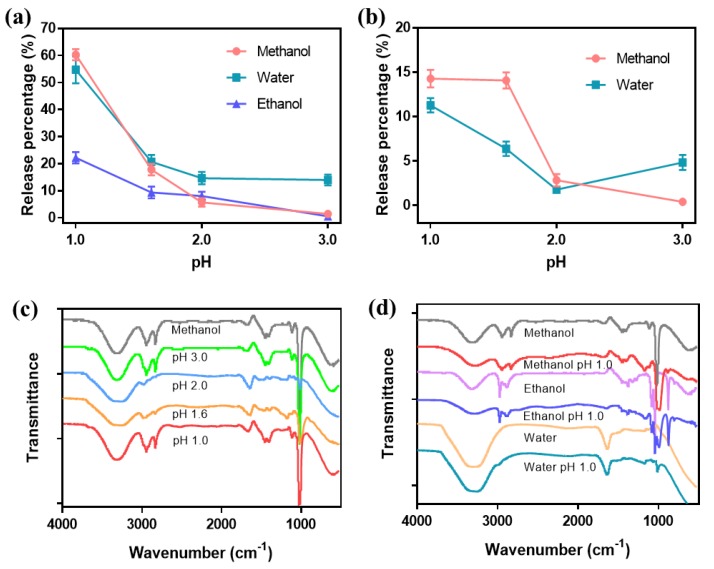
(**a**) Release percentages of anthocyanin from Fe_3_O_4_/anthocyanin magnetic biocomposites for the first time; (**b**) release percentages of anthocyanin from Fe_3_O_4_/anthocyanin magnetic biocomposites for the second time; (**c**) the FTIR spectra of the anthocyanin in methanol solution with different pH values; (**d**) the FTIR spectra of the anthocyanin in different solutions with pH 1.0.

**Figure 6 polymers-11-02077-f006:**
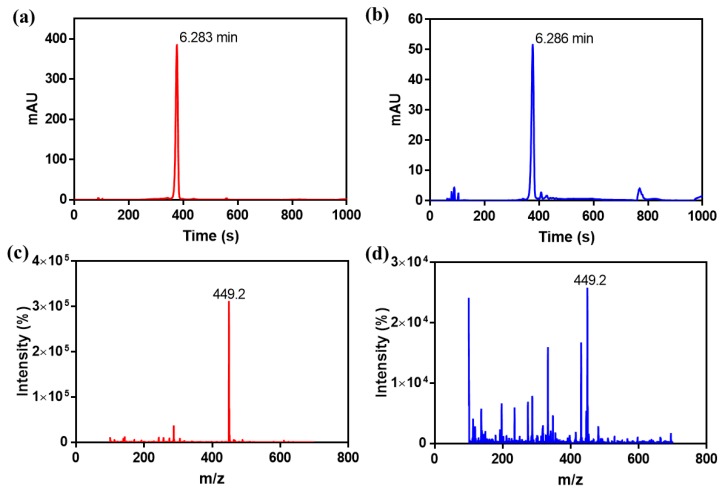
High performance liquid chromatography (HPLC) analysis of anthocyanin before compounding (**a**) and after releasing (**b**). High performance liquid chromatography-mass spectrometry (HPLC-MS) analysis of anthocyanin before compounding (**c**) and after releasing (**d**).

**Table 1 polymers-11-02077-t001:** Zeta potential and particle size of anthocyanin, Fe_3_O_4_ and Fe_3_O_4_/anthocyanin composite.

Materials	Zeta Potential (mV)			Average Value (mV)	Particle Size (nm)			Average Value (nm)
Anthocyanin	+10.90	+11.10	+10.70	+10.90	\	\	\	\
Fe_3_O_4_	−46.38	−42.42	−47.97	−45.59	192.3	203.6	184.1	193.3
Fe_3_O_4_/anthocyanin	−37.25	−38.55	−37.59	−37.70	225.9	230.2	210.5	222.2

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
