# Peer review of "Preparation and pH Controlled Release of Fe3O4/Anthocyanin Magnetic Biocomposites"

_polymers, 2019, doi:10.3390/polym11122077_

Round 1

Reviewer 1 Report

The authors have investigated on controlled release of Fe3O4/anthocyanin composites depending on pH. The preparation of magnetic particle-anthocyanin composite is done by the covalent or physical intermolecular cross-linking, which is rather easy way to stabilize the magnetic particles in aqueous solution. The characterization of the particle and release of anthocyanin was carried out by SEM, DLS, FT-IR, XRD and so on. The conclusion of the experiments sounds good and interesting for the reader in this field. However, there are some issues to be addressed.

1.In Abstract, 
 However, the relatively low stability and bioavailability block the wide and effective applications of anthocyanins. --> This is not complete sentence. There is no verb.

2. The author conducted on the controlled release experiment in room temperature. For the application as drug delievery, the author should conducted this experiment at different temperature (eg. 37C ~ 40C).

These issues should be addressed before publication.

Author Response

Response to Reviewer 1 Comments

The authors have investigated on controlled release of Fe3O4/anthocyanin composites depending on pH. The preparation of magnetic particle-anthocyanin composite is done by the covalent or physical intermolecular cross-linking, which is rather easy way to stabilize the magnetic particles in aqueous solution. The characterization of the particle and release of anthocyanin was carried out by SEM, DLS, FT-IR, XRD and so on. The conclusion of the experiments sounds good and interesting for the reader in this field. However, there are some issues to be addressed.

Point 1:

1.In Abstract,

However, the relatively low stability and bioavailability block the wide and effective applications of anthocyanins. --> This is not complete sentence. There is no verb.

Response 1: We greatly appreciate this suggestion. The sentence was revised in the manuscript as below:

However, the wide and effective applications of anthocyanins have been limited by the relatively low stability and bioavailability.

Point 2: The author conducted on the controlled release experiment in room temperature. For the application as drug delievery, the author should conducted this experiment at different temperature (eg. 37C ~ 40C).

Response 2: We follow this very nice suggestion and conduct the experiment at different temperatures (eg. 30 °C, 35 °C, 40 °C, 45 °C and 50 °C). The results were given in Supporting Information and the comments were revised in the manuscript as below:

Release percentages of anthocyanin from Fe3O4/anthocyanin magnetic biocomposites at different temperatures were also given in Figure S3. As the temperature changed from 30~50 °C, the release percentages remained around 30%, indicating that the magnetic biocomposites could be release stably at 30~50 °C.

Reviewer 2 Report

In my opinion the title is unclear and has to be simplified.

I do not see the reason of Figure 3a. This has no addition to this study.

Figure 6 and 7 do not reach the publishable level in current form and must be improved. They seem like a direct copy/paste from the chromatography software.

The authors claim that the ion at 449 m/z is cyanidin-3-O-glucoside although the molecular weight of this structure is 484.8 based on pubchem. So why and what is the difference? MS/MS fragmentation would be needed to confirm the analyzed structures.

At line 34, authors claim that the developed biocomposite can effectively improve the stability of antocyanins although there is no experiments described on structure stability.

Line 77,  binding behavior and protective effects of Fe3O4…were evaluated. I have not seen any of these in the results

There are no conclusions in the conclusions section, it mixtures methods and results.

Author Response

Response to Reviewer 2 Comments

Point 1: In my opinion the title is unclear and has to be simplified.

Response 1: We follow this very nice suggestion and revise the title in the manuscript as below:

Preparation and pH controlled release of Fe3O4/anthocyanin magnetic biocomposites

Point 2: I do not see the reason of Figure 3a. This has no addition to this study.

Response 2: We really appreciate this suggestion. The morphology of anthocyanin before compounding was shown in Figure 3a. The comments were revised in the manuscript as below:

The morphologies of Fe3O4 nanoparticles and Fe3O4/anthocyanin magnetic biocomposites were characterized by SEM. As shown in Figure 3a, Fe3O4 nanoparticles prepared by the hydrothermal method were well monodispersed with a diameter of approximately 200~300 nm. The surface of Fe3O4 nanoparticles was rough and the structures of the nanoparticles were porous. Compared with the Fe3O4 nanoparticles showed in Figure 3a, the surface of Fe3O4/anthocyanin magnetic biocomposites became smooth, which indicated that anthocyanin has been coated on the surface of the Fe3O4 particles (Figure 3b).

Point 3: Figure 6 and 7 do not reach the publishable level in current form and must be improved. They seem like a direct copy/paste from the chromatography software.

Response 3: We greatly appreciate this suggestion. The new Figure 6 combines Figure 6 and 7 in the manuscript as follows:

Point 4: The authors claim that the ion at 449 m/z is cyanidin-3-O-glucoside although the molecular weight of this structure is 484.8 based on pubchem. So why and what is the difference? MS/MS fragmentation would be needed to confirm the analyzed structures.

Response 4: We greatly appreciate this suggestion. Cyanidin-3-O-glucoside standard was purchased from Shanghai Macklin Biochemical Co., Ltd, China (http://www.macklin.cn/products/C832095). It is a cyanidin-3-O-glucoside chloride. The structure of cyanidin-3-O-glucoside standard was given in Supporting Information (Figure S4). The molecular weight of this structure is 484.8. After preparing the standard solution, the positive ion at 449 m/z is cyanidin-3-O-glucoside while the negative ion at 35.5 m/z is chlorine ion. Therefore, the molecular weight of cyanidin-3-O-glucoside is 449 m/z.

Point 5: At line 34, authors claim that the developed biocomposite can effectively improve the stability of antocyanins although there is no experiments described on structure stability.

Response 5: We really appreciate this suggestion. We hope the biocomposite prepared in this work will effectively improve the thermal stability, light stability and pH stability of antocyanins, but in this work, we mainly study the preparation and release of biocomposite, and we will continue to improve the stability of the biocomposite in our following work. The comment was revised in the manuscript as below:

Therefore, anthocyanin can be effectively adsorbed and released by this magnetic biocomposite.

Point 6: Line 77,  binding behavior and protective effects of Fe3O4…were evaluated. I have not seen any of these in the results

Response 6: We really appreciate this suggestion. The characterizations were carried out by SEM, DLS, FT-IR and XRD to evaluate the successful combination of anthocyanin and Fe3O4.The stability mentioned in point 5 can reflect the protective effect. However, the expression is not so accurate in this article. The comment was revised in the manuscript as below:

A series of performance characterizations were carried out to evaluate the combination of anthocyanin and Fe3O4.

Point 7: There are no conclusions in the conclusions section, it mixtures methods and results.

Response 7: We follow this very nice suggestion and revise the conclusions section in the manuscript as below:

In this paper, an innovative strategy for producing Fe3O4/anthocyanin magnetic biocomposites adopting the magnetic materials as the anthocyanin carriers with a drug release controlled by pH values was prepared. With the introduction of the hydroxyl and electrostatic adsorption charges, the Fe3O4/anthocyanin magnetic biocomposites were supposed to be coordinated on the physical intermolecular or covalent cross-linking between Fe3O4 and anthocyanin. This strategy provides a feasible method for preparing Fe3O4/anthocyanin magnetic biocomposites with designated chemical groups. Magnetic biocomposites have an average size about 222 nm. The primary release percentage of anthocyanin in acid methanol was 60.9%. The preparation method of this work is simple, and the prepared biocomposites have good biocompatibility and could be controlled released. The biocomposites could be released at the target position by magnetic guide. Therefore, the methodology of Fe3O4/anthocyanin magnetic biocomposites presented in this work provide an effective way to combine, adsorb and release anthocyanin, which has great potential value in the applications of medical, food and sensing.

Round 2

Reviewer 2 Report

The first sentence of the conclusion is still confusing. If you change the title of the manuscript it has to be changed in the supplementary as well.

Author Response

-The first sentence of the conclusion is still confusing. If you change the title of the manuscript it has to be changed in the supplementary as well.

Response: We greatly appreciate this suggestion. The first sentence of the conclusion was revised in the manuscript as below:

In this paper, an innovative strategy for the preparation of Fe3O4/anthocyanin magnetic biocomposites was proposed. Anthocyanin could be controlled released from the biocomposites by pH values.

The title has been changed in the supplementary as below:

Preparation and pH controlled release of Fe3O4/anthocyanin magnetic biocomposites